# OncoPan^®^: An NGS-Based Screening Methodology to Identify Molecular Markers for Therapy and Risk Assessment in Pancreatic Ductal Adenocarcinoma

**DOI:** 10.3390/biomedicines10051208

**Published:** 2022-05-23

**Authors:** Maria Grazia Tibiletti, Ileana Carnevali, Valeria Pensotti, Anna Maria Chiaravalli, Sofia Facchi, Sara Volorio, Frederique Mariette, Paolo Mariani, Stefano Fortuzzi, Marco Alessandro Pierotti, Fausto Sessa

**Affiliations:** 1Unit of Pathology, Ospedale di Circolo, ASST-Sette Laghi, 21000 Varese, Italy; annamaria.chiaravalli@asst-settelaghi.it (A.M.C.); fausto.sessa@uninsubria.it (F.S.); 2Research Center for the Study of Hereditary and Familial Tumors, University of Insubria, 21100 Varese, Italy; sofia.facchi@uninsubria.it; 3Cogentech Società Benefit S.R.L., 20139 Milan, Italy; valeria.pensotti@cogentech.it (V.P.); sara.volorio@cogentech.it (S.V.); frederique.mariette@cogentech.it (F.M.); paolo.mariani@cogentech.it (P.M.); stefano.fortuzzi@cogentech.it (S.F.); marco.pierotti@cogentech.it (M.A.P.); 4Department of Medicine and Surgery, University of Insubria, 21100 Varese, Italy; 5Istituto Fondazione di Oncologia Molecolare (IFOM), 20139 Milan, Italy

**Keywords:** pancreatic adenocarcinoma, OncoPan^®^ NGS analysis, *HER2* amplification, *HR* genes mutations, target therapy, risk assessment

## Abstract

Pancreatic cancer has a high morbidity and mortality with the majority being PC ductal adenocarcinomas (PDAC). Whole genome sequencing provides a wide description of genomic events involved in pancreatic carcinogenesis and identifies putative biomarkers for new therapeutic approaches. However, currently, there are no approved treatments targeting driver mutations in PDAC that could produce clinical benefit for PDAC patients. A proportion of 5–10% of PDAC have a hereditary origin involving germline variants of homologous recombination genes, such as Mismatch Repair (MMR), *STK11* and *CDKN2A* genes. Very recently, *BRCA* genes have been demonstrated as a useful biomarker for PARP-inhibitor (PARPi) treatments. In this study, a series of 21 FFPE PDACs were analyzed using OncoPan^®^, a strategic next-generation sequencing (NGS) panel of 37 genes, useful for identification of therapeutic targets and inherited cancer syndromes. Interestingly, this approach, successful also on minute pancreatic specimens, identified biomarkers for personalized therapy in five PDAC patients, including two cases with *HER2* amplification and three cases with mutations in *HR* genes (*BRCA1*, *BRCA2* and *FANCM*) and potentially eligible to PARPi therapy. Molecular analysis on normal tissue identified one PDAC patient as a carrier of a germline *BRCA1* pathogenetic variant and, noteworthy, this patient was a member of a family affected by inherited breast and ovarian cancer conditions. This study demonstrates that the OncoPan^®^ NGS-based panel constitutes an efficient methodology for the molecular profiling of PDAC, suitable for identifying molecular markers both for therapy and risk assessment. Our data demonstrate the feasibility and utility of these NGS analysis in the routine setting of PDAC molecular characterization.

## 1. Introduction

Pancreatic cancer (PC) has high morbidity and mortality [1]. It is the fourth most common cause of cancer death in Western societies and is projected to be the second leading cause within a decade. For these reasons, therapeutic and prevention strategies are a priority for increasing patient’s care and survival. The majority of PC (94%) are pancreatic ductal adenocarcinomas (PDAC) that develop in the exocrine tissue of the pancreas. PDAC arises in the sporadic setting, and/or associated with a known inherited cancer syndrome. Whole genomic sequencing of PDAC provides a wide description of genomic events involved in pancreatic carcinogenesis and identifies putative biomarkers for new therapeutic approaches [2]. There are currently no approved treatments that target driver mutations in PDAC, such as *KRAS*, *TP53*, *CDKN2A* and *SMAD4*, that could produce clinical benefits for PDAC patients.

A proportion of 5–10% of PDAC have a hereditary origin showing an identifiable germline mutation. The most frequent cancer genes involved in susceptibility of PDAC include *BRCA1* and *BRCA2*, *CDKN2A*, *STK11* and MMR genes [3]. Even if methods for prevention or early detection of pancreatic cancer have limited utility [4], PDAC may be a useful sentinel cancer for the identification of carriers of pathogenic variants in well defined cancer syndromes. It is indeed relevant that at-risk relatives can benefit from surveillance, medical and surgical strategies for prevention, risk reduction or early detection.

Although supported by a low level of evidence, exquisite sensitivity of *BRCA1* and *BRCA2* mutant tumors to platinum compounds has been validated in multiple preclinical and clinical studies [5,6].

Very recently, the international randomized, placebo-controlled phase III POLO trial demonstrated that maintenance Olaparib therapy significantly prolongs progression-free survival in metastatic PDAC patients carrier of pathogenic *BRCA1* and *BRCA2* variants whose disease had not progressed during first-line platinum-based chemotherapy [7].

Again, solid tumors with MMR deficiency often respond to immunotherapy [8] and MMR defects are a hallmark molecular feature of Lynch syndrome related cancers.

In summary the availability of targeted treatments increases the utility of genetic testing for pancreatic cancer patients themselves, in addition to the prevention and screening benefits for relatives.

Here we reported results of a target NGS analysis of 21 FFPE PDAC samples using a strategic panel of 37 genes, including possible targets for therapies and cancer susceptibility genes involved in PDAC. The aim of this study was to support the utility of OncoPan^®^ in clinical practice for both therapy and prevention in PDAC patients.

## 2. Materials and Methods

### 2.1. Patients and Tissues

Twenty-one primary PDACs surgically resected between 2011 and 2019 were retrieved from the files of Anatomic Unit of ASST-Settelaghi of Varese. The series included 6 males and 15 females with the mean age of onset of 66.6 (range 50–76) and 68.4 (range 51–82), respectively, for males and females. The sites of the tumors were head in 13 cases, head-tail in 1 case, body-tail in 4 cases, tail in 1 case and 1 choledocus. Histological grade was moderately differentiated in 13 cases and poorly differentiated in 8 cases, according to the 2019 WHO classification of tumors of the digestive system.

All samples analyzed with OncoPan^®^ were retrospectively acquired and restricted to primary operable non-pretreated PDAC. All patients included in this study had died, however, the overall survival data are not available. Once the diagnosis of PDAC was checked, representative sections were reviewed independently by expert pathologist with specific expertise in pancreatic diseases. Only Formalin Fixed Embedded Paraffin (FFPE) blocks with a tumor cellularity of at least 30–50% were selected and used for DNA extraction.

Tissue from 6–8 slides (8 μm) of each FFPE block were scraped off the slides. Tumor DNA was extracted through the Gene Read DNA FFPE kit (Qiagen, Hilden, Germany) according to the protocol instructions. In 3/21 cases, it was necessary to concentrate the obtained DNA (through vacuum concentrator), as it was too diluted (3–5 ng/μL) to use it in the NGS protocol, which required DNA with a concentration of at least 15–20 ng/μL. In one case, it was necessary to pool the DNA from two independent extractions (both from 6–8 slides). In another three cases, the DNA obtained was very little, but fortunately different FFPE blocks were available for DNA extraction and subsequent NGS analysis.

For germline analysis, DNA extracted from non-neoplastic tissue, isolated from each sample after pathology review was used.

This study was conducted in accordance with recognized ethical guidelines (Declaration of Helsinki), and the use of tumor samples was approved by the local ethics committee of the ATS Insubria (Study 199, 2019 approved on 19 May 2020). All analyses were performed as summarized in Figure 1.

### 2.2. NGS Analysis

For NGS analysis, a custom hybrid capture-base panel was designed (SureDesign 6.9 application, Agilent Technologies, Santa Clara, CA, USA) and named OncoPan^®^. It can be used to detect single-nucleotide variants (SNVs), small insertions/deletions (INDELs) and copy-number variations (CNVs). OncoPan^®^ was implemented to be used in the routinely clinical practice in a cancer genetic laboratory: it was designed to find both gene variants that are possible targets for therapies and germline variants in cancer susceptibility genes, for the benefit of at-risk relatives. The analysis covers at least all the coding exons of the 37 genes and two pseudogenes listed in Appendix A. *EPCAM* analysis is restricted to the detection of copy number variations, *CDK4* analysis is restricted to exon 2.

About 150–200 ng of dsDNA, according to Qubit dsDNA HS assay kits fluorimetric quantification, were sheared by Sure Select Enzymatic Fragmentation kit (Agilent Technologies Inc., Santa Clara, CA, USA). NGS libraries were created using Sure Select XT Low input Custom library kit (Agilent Technologies Inc.) and sequencing was performed on MiSeqDX (Illumina Inc., San Diego, CA, USA) through 2 × 150 bp paired-end module. Data were collected by LRM v3.1 software (Illumina, San Diego, CA, USA), using the ‘FastQ only’ workflow. The run quality was evaluated by Illumina Sequencing Analysis Viewer v.1.11.1 (Illumina, San Diego, CA, USA), while bioinformatics pipeline for the creation of SNV and CNV calls was developed in house, in collaboration with the enGenome Software Company (Pavia, Italy). SNVs and INDELs are identified and genotyped versus Human hg19 genome by VarDict, FreeBayes, Mutect2 and Scalpel software, producing .vcf files that are annotated by both an eVai tool (enGenome, Pavia, Italy) and an internal laboratory database.

CNVs (in terms of single-exon or multi-exon deletions or duplications) are called via PureCN and CNVkit coverage-based algorithms. CNV procedure generates a tab delimited .tsv file, containing the list of CNVs identified by at least one of the tools used for CNV calling.

The data analysis showed that all the exons were covered by at least 50 reads; with a minimum sample’s coverage depth of 57× and a maximum sample’s coverage depth of 345×: we achieved an average sample read depth of 160×. The minimum number of reads useful to call a variant was 10. Six low frequency mutant alleles (less than 10%), although covered by only 5 to 9 reads, were included anyway in list of variants identified, but after visual inspection (by Integrative Genomics Viewer (IGV), Broad Institute, CA, USA) of the corresponding sequence in all the samples, in order to verify the nucleotide-specific noise level.

Only 4 samples out of 21 (PA8, PA9, PA18 and PA20) were below our standard parameters, with an average coverage depth between 20× and 45×.

### 2.3. Sanger Sequencing

Sanger sequencing was performed on DNA extracted from non-neoplastic tissues to investigate if the variants detected by NGS were of germline origin or not. Oligonucleotides specific for each variant were designed and the corresponding DNA regions were amplified at the annealing temperature of 60 °C, with the AmpliTaq Gold kit (Applied Biosystems; Thermo Fisher Scientific, Inc., Waltham, MA, USA). Sequencing was performed on purified PCR products by using the BigDye^®^ Terminator v.3.1 Cycle Sequencing kit (Thermo Fisher Scientific, Inc., Waltham, MA, USA) and run on the 3500 Dx Genetic Analyzer (Thermo Fisher Scientific Inc.) The obtained data were analyzed by Mutation Surveyor^®^ Software v5.1.2 (SoftGenetics, LLC, State College, PA, USA).

### 2.4. Multiplex Ligation-Dependent Probe Amplification (MLPA)

The analysis of large deletions/duplications in the *BRCA1* gene was carried out on DNA extracted from non-neoplastic tissues from case PA3 with the *BRCA1* SALSA MLPA KIT-P087 (D1) probemix (MRC-Holland, Amsterdam, The Netherlands), following the manufacturer’s instructions. MLPA products were run on the 3730Xl DNA Analyzer (Applied Biosystems; Thermo Fisher Scientific, Inc.) with the fragment analysis module. The results were analyzed through the Gene Marker Software v3.0.1 (SoftGenetics, LLC, State College, PA, USA).

### 2.5. Variant Classification

The sequence variant nomenclature followed the Human Genome Variation Society (HGVS) guidelines v.20.05 [9]. The final classification as pathogenic (class 5), likely pathogenic (class 4), uncertain significance or VUS (class 3) followed to the American College of Medical Genetics and Genomics Standards and Guidelines for the Interpretation of Sequence Variants [10]. Likely benign (class 2) and benign (class 1) variants were not reported.

### 2.6. FISH Analysis

Interphase FISH analysis was performed on 3–4-μm sections used for conventional histological examination according to the guidelines of the European Cytogeneticists Association (European Cytogenetic Guidelines: www.e-c-a.eu (Guidelines, NEWSLETTER No. 29 January 2012)). The experiments were carried out as previously described [11]. The Pathvision (Vysis) probe that simultaneously hybridizes *HER2* gene (red labeled) and the centromere of chromosome 17 (green labeled) was used. FISH analysis was performed using a Bioview (Abbott, Chicago, IL, USA) and tissue matching procedure on more than 200 interphase nuclei from 5 to 8 separate areas of the tumor selected for well-preserved cellular and nuclear morphology by 2 independent operators. Only experiments with 90% hybridization efficiency were considered. *HER2* amplification gene was evaluated when the ratio between gene signals and centromeres were >2 as reported by ASCO guidelines [12] and having regard to genetic heterogeneity for the presence of small clones with gene amplification.

### 2.7. Immunohistochemical Analysis

The immunohistochemical study was performed on 3 μm FFPE consecutive sections. All the immunohistochemical staining was automatically processed on Benchmark autostainer, according to routine protocols. In particular, *HER2* expression was detected with UltraView DAB Detection Kit (Ventana Medical System, Oro Valley, AZ, USA) and Anti-Human c-erbB-2 Oncoprotein polyclonal antibody (A0485, DAKO, Agilent technologies, Santa Clara, CA, USA) after 36 min of antigen retrieval with CC1 solution. *HER2* over-expression was evaluated according to guidelines for Gastroesophageal adenocarcinoma and Rushoff-Hofmann scoring system [13,14].

MMR protein expression was analyzed using specific monoclonal antibodies against MSH2, MSH6, MSH3, MLH1 and PMS2 proteins as previously described [15]. A case was considered defective for a protein when all the tumor nuclei failed to react with the specific antibody. Intact nuclear staining of adjacent normal epithelial cells, lymphocytes and fibroblasts were used as internal positive control.

## 3. Results

### 3.1. NGS Results

Targeted NGS testing detected at least one variant in each of the 21 PDAC with a median of 3.5 variants per tumor (range 1–27). The variants identified were 82 overall, including 41 pathogenic (class 4 and 5) and 40 variants with unknown clinical significance (class 3). Allelic frequency of all variants ranged from 2 to 56%. Figure 2 shows the distribution of variants identified in each gene of the panel. The genes are listed from the most mutated (top) to the least mutated (bottom), in decreasing order in our cohort of samples. SNVs identified specifically in *CHEK2* or *PMS2* are located in exons not homologous to their pseudogenes. Moreover, large deletions and duplications (CNVs) have not been taken into account when detected in samples with an average sample read depth under 75× (8/21 cases) or in genes with pseudogenes. The complete list of variants is shown in Appendix A.

Pathogenic variants of *KRAS* and *TP53* were the most frequent, as described in the literature [16]. They were present, respectively, in 76% and 57% of PDACs. All *TP53* variants were classified as pathogenic and reported in other tumors. All but one *KRAS* variant was classified as pathogenic and reported in other tumors. *SMAD4* gene was involved in 5 out of 21 cases (4 pathogenic variants and 1 VUS).

Mutations in cell cycle genes were the most common actionable alterations and, among them, the most frequently mutated gene was *CDKN2A*, observed in 3 out of 21 PDAC.

Pathogenic variants of DNA repair genes were detected in three cases including, respectively, *BRCA1*, *BRCA2* and *FANCM*. Variants of MMR genes were observed in 7 PDACs including two cases with class 4 and 5 variants and 5 PDAC with class 3 variants. Moreover, two additional VUS were detected in *PMS2*, but it could not be excluded that they were located in the corresponding pseudogene *PMS2CL*. Three cases of those involving MMR genes showed more than two variants, as described in Table 1.

Moreover, two not previously described point mutations of the *HER2* gene were also observed, while no variants in *APC* and *STK11* genes were detected.

Interestingly, one variant in the *RAD51D* gene, detected in a single PDAC (PA10 case), is a benign variant, quite frequent in the population (gnomAD NFE: 1.68%). Functional study suggests that this variant affects *RAD51D* functions and protein interactions, by increasing cellular resistance to DNA damaging agents (chemoresistant). It seems to contribute to telomere dysfunction by conferring cellular proliferation and decreasing the interaction with *RAD51C*. It confers increased cisplatin resistance and cell growth phenotypes in human breast carcinoma cell lines with a mutant *TP53* gene [17].

Due to the presence of pseudogenes, it was not possible to ascertain if variants identified in most of the exons of *CHEK2* or *PMS2* were actually in the genes or in the corresponding pseudogenes. In fact, long range PCR is only usable efficiently on intact DNA, such as the one extracted from blood or fresh tissues.

Overall, in agreement with chromosome instability of PDAC, an increased copy number of several genes was observed (see Figure 2). Regarding actionable genes, CNV analysis revealed high copy number of *HER2* in PA4 case (Figure 3) and 4 *HER2* copies in four additional cases.

### 3.2. Germline Results (Sanger Sequencing and MLPA)

In order to ascertain the germline condition of actionable variants, Sanger sequencing of eight PDAC was performed on DNA extracted from non-neoplastic tissue, as blood samples were not available. The investigated variants are highlighted in grey in Appendix A. In detail, six pathogenic variants of *CDKN2A*, *BRCA2*, *FANCM* and *MSH2* and four variants of uncertain significance (VUS) of *CDKN2A*, *BRCA2* and *MSH2* were investigated by Sanger sequencing, but none of them were present in the non-neoplastic tissue of the corresponding PDAC.

Instead, the benign variant in the *RAD51* gene was confirmed to have a germline origin.

Notably, MLPA analysis on the DNA from the non-neoplastic tissue from PA3 case was able to highlight the pathogenic deletion of exon 16 of *BRCA1*, previously identified in the tumor sample (Figure 4). This variant had a germline origin, in agreement with personal and family history of breast cancer of this PDAC patient. The pedigree of this family affected by breast and ovarian cancer syndrome (ORPHA 145) is described in Figure 4.

### 3.3. FISH Results

FISH analysis was performed on five PDAC showing more than four copy numbers of the *HER2* gene (PA1, PA3, PA4, PA15, PA21 in Table 2 and on two additional PDAC showing point mutation of *HER2* gene (PA8 and PA20 in Table 2) at NGS analysis. Case PA4 revealed high levels of *HER2* amplification showing a ratio between *HER2* genes and chromosome 17 centromere more than 2.00 (average of *HER2*/cell: 15.33; average of 17 centromere: 4.00) (Figure 3). This case was also investigated for *TOP2A* gene located near to *HER2* gene, but the amplification involved only *HER2* gene (see Appendix A).

Three PDACs showing four *HER2* copy number at NGS analysis were negative for *HER2* amplification, while the remaining PDAC revealed 20% of cells with *HER2* amplification and was classified as a heterogeneous tumor for *HER2* amplification (see Table 2).

FISH analysis was also performed in two PDAC, showing point mutations located at the end of the protein tyrosine kinase domain of the *HER2* gene, but no amplification was observed. These variants had not been previously described and thus there is no evidence available to consider them activating mutations. The variant p.(Gly909Ser), located in the last base of exon 22, results deleterious in silico splicing predictions as it strongly weakens splicing donor site, while the variant p.(Arg896Profs*8) introduces a premature termination codon (see Appendix A).

### 3.4. IHC Results

*HER2* IHC expression was investigated on seven cases including five PDACs with more than four *HER2* copy numbers and two PDACs showing mutation of the *HER2* gene (in Table 2).

In agreement with the presence of *HER2* gene amplification detected by FISH analysis, the case PA4 demonstrates 3+ score positivity, showing strong complete or basolateral membrane immunoreactivity in 90% of tumor cells (Figure 3B; Table 2)

Interestingly, case PA1 displayed an equivocal *HER2* immunohistochemical pattern of expression (2+ score), showing weak–moderate basolateral membrane immunoreactivity in 10% of tumor cells; FISH analysis revealed a heterogeneous pattern for HER2 amplification. No *HER2* immunoreactivity was observed in all the other cases. Appendix A shows immunohistochemical expression and FISH results of cases PA4, PA1 and PA3.

MMR gene immunohistochemical analysis was performed for all the PDACs showing both VUS and/or pathogenic variants of *MSH2*, *MSH3*, *MSH6* and *PMS2* genes (cases PA1, PA7, PA9, PA12, PA13, PA18 and PA19). Appendix A shows an example of MMR genes expression in PA12. PDAC nuclear immunoreactivity for MMR proteins was observed in all the cases with the exception of case PA13. In this case, a heterogeneous pattern of *PMS2* expression was observed: next to the areas with normal nuclear immunoreactivity there were areas where tumor cells showed absence of *PMS2* nuclear expression along with cytoplasmatic accumulation of the protein. In the same case, a reduced expression of *MSH3* was observed too (Table 1). Microsatellite instability was investigated in all MMR mutated PDAC but no instability was detected (data not shown).

## 4. Discussion

This study aimed to investigate the mutational spectrum of PDAC, in order to support the utility in clinical practice of a previously designed and validated strategic panel, named OncoPan^®^, for both therapy and risk assessment in PDAC patients, through the analysis of their FFPE tumor tissues.

The OncoPan^®^ panel of 37 genes used in this study includes evaluation of single-nucleotide variants, short insertions and deletions, copy number amplifications and deletions of a series of actionable genes. This approach was successful on FFPE tissue and also when minute specimens or samples with low tumor content were available.

Although there was a restricted number of PDAC samples, all belonging to a single Institution in Italy, our results resemble large scale studies, with a prevalence of *KRAS* and *TP53* somatic mutations in agreement with other studies as reported by TCGA study [2,18,19,20] and by recent publication of Zhang et al. [21]. Pathogenic somatic variants in *SMAD4*, *CDKN2A*, *BRCA2*, *FANCM*, MSH6 and *MSH2* were also identified, with allelic frequency ranging from 2 to 56% (see Appendix A). Most of these genes, except FANCM, have already been identified in larger studies as mutated in PDAC tumors [2,19].

Since the overall survival of PDAC patients included in this study is not known, somatic and germline identified variants should not be correlate with patient’s prognosis. However, the identified variants will be useful to explore therapeutic markers for this disease.

Currently, there are no approved treatments that target driver mutations such as *KRAS*, *TP53*, *CDKN2A* and *SMAD4* that could produce clinical benefit for PDAC patients; however, several clinical studies are ongoing including phase I and II studies (www.cliniclatrial.gov (accessed on 26 January 2022)).

Interestingly, alterations of HR pathway including pathogenic variants of *BRCA1*, *BRCA2* and *FANCM* genes were identified in three PDAC samples and this finding had important clinical implications as potential biomarker of therapeutic vulnerability to DNA damage agents, such as platinum and PARP inhibitors. Clinical trials in patients with *BRCA*-mutated PDAC have reported positive responses to PARP inhibitors [7,22] and novel prospective trials to include PDAC patients also with *BRCA**ness* properties or HR deficiency could potentially result in better therapeutic approaches for these patients, more effective treatment outcomes, longer survival and subsequently replace current standard of care.

An interesting variant in the *RAD51D* gene was detected in a single PDAC (PA10 case); this is a benign variant, but seems to confer increased cisplatin resistance, whenever it is in the presence of a mutated *TP53*. This finding would have been helpful in the choice of the therapy, as the patient’s tumor DNA (PA10 case) harbors a *TP53* pathogenic variant.

It has also been documented that a small proportion (1–2%) of PDAC [23] has DNA mismatch repair defects (MMR) and can potentially be treated with immune checkpoint blockade therapy [8]. In our study, two cases showed pathogenic *MSH2* and *MSH6* variants; however, none of these revealed loss of *MSH2* and *MSH6* proteins expression and/or microsatellite instability suggesting tumoral heterozygosity of MMR pathogenic variants. It is well known that pembrolizumab was approved for adult patients with loss of *MMR* genes expression or microsatellite unstable cancers, a hallmark molecular feature of Lynch syndrome related cancers. However, at the moment, data about immunotherapy performance on cancers with heterozygous MMR variants are not available.

NGS analysis also revealed *HER2* alterations, including potentially deleterious variants in two cases, high copy number in one case and four copy number in four cases. FISH and IHC analyses of these cases revealed *HER2* gene amplification in only two cases including one PDAC showing a high level of amplification in all neoplastic cells and one PDAC showing a heterogeneous pattern of *HER2* amplification (20% of amplified cells) in one case. These results are clinically relevant because several clinical studies using trastuzumab in pancreatic cancer are ongoing (www.clinicaltrials.gov (accessed on 26 January 2022)) and these two patients would probably have benefited from an off label trastuzumab therapy. Very recently, Hirokawa [24] et al. described an effective trastuzumab treatment in patient with heterotopic pancreatic adenocarcinoma *HER2* positive.

Even if our study is limited to a little subset of PDAC and germline, analysis of variants identified on tumoral tissues was possible only in eight cases, it is noteworthy that NGS approach on tumor tissue let us to identify the exon 16 deletion of *BRCA1* in the germline setting of a PDAC patient. This patient was a member of a well-known family affected by inherited breast and ovarian cancer syndrome (ORPHA 145) (see Figure 4). This is a relevant result because it demonstrates that NGS on PDAC tumor tissue is a useful approach to reveal an inherited cancer condition even when family history or guidelines may not predict mutational status, and if PDAC patients died and are not available for germline analysis. Recently, Peretti et al. [25] published an epidemiological study of an Italian patients’ cohort and demonstrated that *BRCA* germline variants incidence is higher than expected in real-life series of PDAC patients. In some families PDAC often represents a sentinel cancer for hidden predisposition in the families: in a large study of 225 PDAC Italian patients, Ghiorzo et al. (2012) [26] demonstrated that there was a strong correlation between familial pancreatic cancer and the *CDKN2A* mutation. In our study, three pathogenic variants of the *CDKN2A* gene were detected; however, these variants were not observed in the germline setting.

Our findings also demonstrate the presence of MMR variants in PDACs, but none of the identified pathogenic variants were demonstrated to be of germline origin; however, it is well known from literature that 1.3% of probands carry a pathogenic allele in one of the Lynch syndrome genes [27].

In summary, somatic genetic analysis is a useful method to identify patients affected by inherited cancer conditions and their asymptomatic at-risk relatives who could benefit from specific and efficient surveillance measures.

## 5. Conclusions

In conclusion, the OncoPan^®^ panel of 37 genes, designed for molecular profiling of PDAC, is an efficient tool for both therapy choice and risk assessment and our data demonstrate that this NGS analysis is robust and performant to be applied in the routine setting of PDAC diagnosis also when the availability of tumoral tissue samples is limited.

## Figures and Tables

**Figure 1 biomedicines-10-01208-f001:**
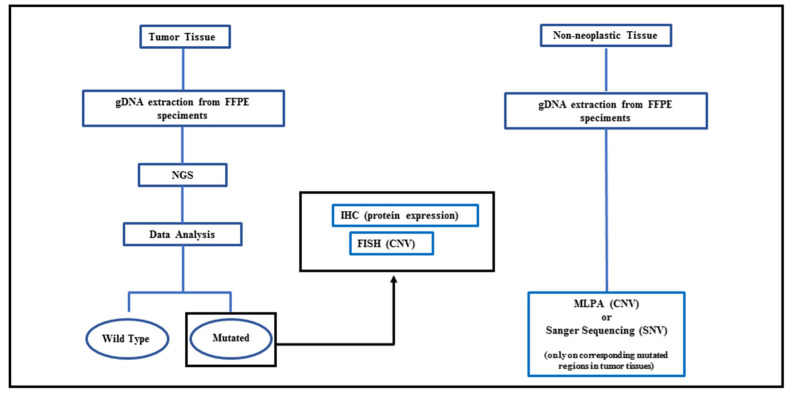
Flow-chart of overall methodology used in this study.

**Figure 2 biomedicines-10-01208-f002:**
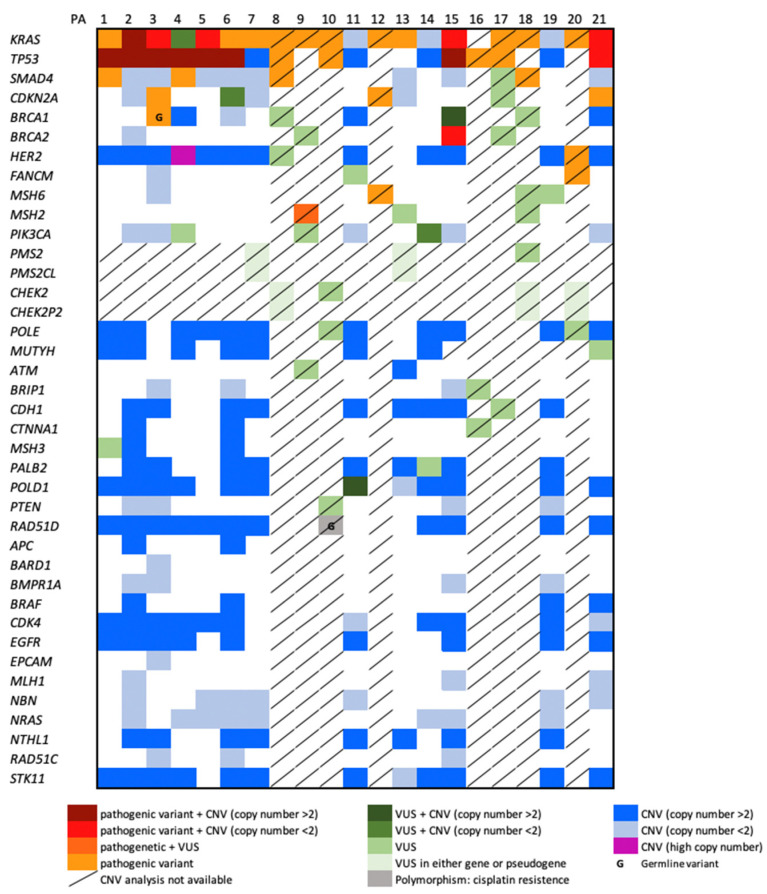
Mutational landscape of investigated genes. The panel shows single nucleotide variants (SNV), ins/del and also copy number variations (CNV) of the entire gene (>2 or <2). Germline variants are indicated with “G”. Cases or genes for which CNV analysis was not available or unreliable for the presence of pseudogenes are indicated by a slash.

**Figure 3 biomedicines-10-01208-f003:**
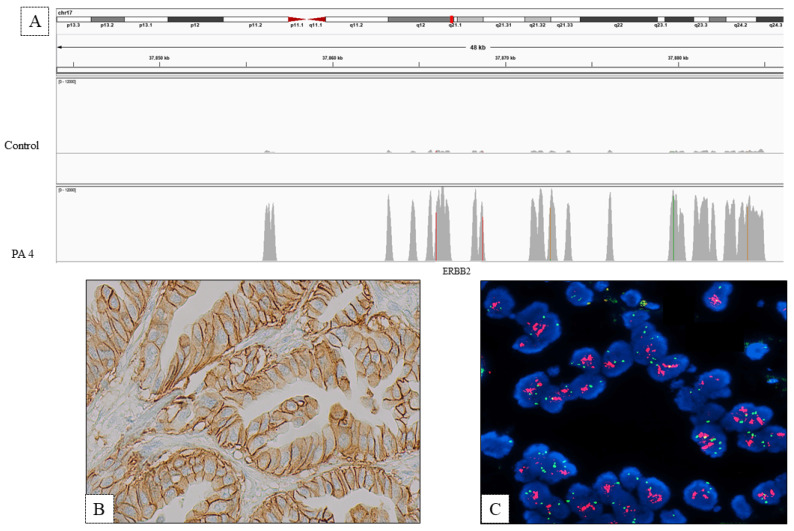
(**A**) *HER2* copy number analysis by integrative genomics viewer (IGV): coverage comparison between PA4 sample and a two copies control (read depth range min 0 _max 12,000). (**B**) Immunohistochemical *HER2* expression. (**C**) FISH analysis showing *HER2* amplification (red signals) respect to chromosome 17 centromere (green signals).

**Figure 4 biomedicines-10-01208-f004:**
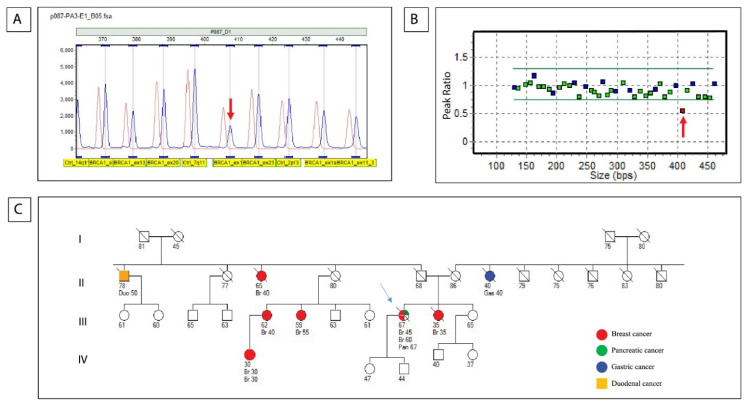
Confirmation of the presence of the germline deletion of exon 16 of *BRCA1* in PA3 non-neoplastic tissue, using the SALSA^®^ MLPA^®^ P087-D1 probe mix. Raw data indicating a deletion of the *BRCA1* probe sized 407 kb (**A**,**B**); two different graphical representation of the results using GeneMarker^®^ software from SoftGenetics v.3.0.1. The arrows indicate the deleted probes. The expected result for a heterozygous deletion is 0.40 < DQ < 0.65. (**C**) Family history of PDAC patient carrier of germline *BRCA1* exon 16 deletion indicated by light blue arrow.

**Table 1 biomedicines-10-01208-t001:** Immunohistochemical expression of MMR proteins in PDACs with variants of MMR genes.

Case	NGS Variant	HGVS Class	Immunohistochemical Expression				
			*MSH2*	*MSH6*	*MSH3*	*MLH1*	*PMS2*
PA1	*MSH3*: c.1088C>A p.(Thr363Asn)	3	POS	POS	POS	POS	POS
PA7	*PMS2*: c.2380C>T p.(Pro794Ser)	3	POS	POS	POS	POS	POS
PA9	*MSH2*: c.301G>T p.(Glu101 *)	5	POS	POS	POS	POS	POS
	*MSH2:* c.2703A>C p.(Glu901Asp)	3					
PA12	*MSH6:* c.3126_3172 + 38del p.?	4	POS	POS	POS	POS	POS
PA13	*MSH2:* c.435T>G p.(Ile145Met)	3	POS	POS	POS/NEG	POS	POS/NEG *
	*PMS2*: c.1148A>T p.(Asn383Ile)	3					
PA18	*MSH2*: c.4_78del p.(Ala2_Met26del)	3	POS	POS	POS	POS	POS
	*PMS2:* c.1004A>G p.(Asn335Ser)	3					
	*MSH6*: c.1957_2010del p.(Val653_Gly670del)	3					
PA19	*MSH6*: c.866G>C p.(Gly289Ala)	3	POS	POS	POS	POS	POS

* Heterogeneous pattern of PMS2 expression (see Section 3).

**Table 2 biomedicines-10-01208-t002:** *HER2* status in 5 PDCAs.

Case	NGS Copy Number	FISH Copy Number	Ratio *HER2*/CEN 17	IHC Expression (% of Tumor Cell)	IHC Expression (Score)
PA1	4	4.92	1.26 *	10	2+
PA3	4	3.80	1.31	NEG	0
PA4	36	15.33	3.83	90	3+
PA15	4	2.54	1.11	NEG	0
PA21	4	2.77	1.13	NEG	0
PA20	**			NEG	0
PA8	**			NEG	0

* These cases had a small cell population (20% of cells) with *HER2* amplification; ** these PDAC had *HER2* variant at NGS analysis.

## Data Availability

Not applicable.

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
