# Peer review of "OncoPan®: An NGS-Based Screening Methodology to Identify Molecular Markers for Therapy and Risk Assessment in Pancreatic Ductal Adenocarcinoma"

_biomedicines, 2022, doi:10.3390/biomedicines10051208_

Round 1

Reviewer 1 Report

Most of my comments have been addressed in the revised manuscript and it is largely improved over the original version. I would like to recommend its publication.

It would be better if the authors could also address my following comments:

  1. The resolution of Figures 2 and 4 is still terrible. I highly recommended using Adobe Illustrator software to generate vector graphics for all the figures.

  1. Although the response said “the prognosis was not a goal of this work”, the authors only want to “develop an easy and reproducible test to evaluate genes involved in prognosis and therapeutic choices”. I still highly recommended analyzing the survival rate of PDAC patients for each identified variant from public datasets, such as TCGA, which can indicate the importance of identified variants in PDAC.

Reviewer 2 Report

The study investigates the utility of a 37gene NGS panel in screening PDACs for genomic alterations of prognostic and therapeutic potential. A relatively small sub-set of 21 samples were tested. Paired germlines were tested for 8 samples. In addition to NGS, MLPA, Sanger sequencing, FISH and IHC studies were used as additional platforms to study as needed. 

The study identified mutations, which have been characteristically identified earlier like, KRAS, TP53 etc. Less frequent mutations were found in genes like BRCA1/2, FANCM. Additional, albeit, less frequent alterations in genes were also detected ( HER2,  RAD51D, PMS2 etc). Paired germline analysis also identified  an exon 16 deletion in BRCA1. As the cohort of cases is small, it will take a bigger sample set to see if the mutations in these additional, low  occurence genes are characteristic of PDACs.  The following points need to be explained and added to improve the clarity of the manuscript

  1. What was the minimum sequencing depth necessary to consider the sequencing was adequate? Please add that to methods. Looks like average depth was 160X. This seems to be low for detection of somatic mutations. for e.g when 2% mutation is detected, it will have mutations in 3 reads total, which may be a low-level artifact. Please clarify what were the minimum qualifications (total depth, minimum mutant reads needed) to accept a mutation
  2. Fig 1: Please clarify if IHC, FISH and MLPA was performed on both WildType and Mutated samples. The diagram appears to suggest that it these were performed only in Mutated samples were tested by IHC and FISH. Modify the figure to give it more clarity

Author Response

This manuscript is a resubmission of an earlier submission. The following is a list of the peer review reports and author responses from that submission.

Round 1

Reviewer 1 Report

This manuscript is written for NGS Panel in Clinical Practice for Therapy and Risk Assessment of Ductal Pancreas Carcinomas. But this manuscript was sometimes reported and not very remarkable. The authors should mention strongly the new point in the paper/abstract. Although the authors performed the NGS analysis, Sanger sequencing MLPA method, FISH, and immunohistochemical analysis, it is not always necessary and not practical. To do multiple method might be spoiled the benefits of the NGS (If the authors have shown the utility of the NGS analysis, you should modify the abstract.). 

The authors described "Hystological (spelling errors?) grade was G2 in 75 13 cases and G3 in 8 cases" in "Materials and Methods", and what dose is mean? Does it have something to do with? And what is G2/G3? The pancreatic adenocarcinoma is typically classified well/moderately/poorly in the histological findings.

Author Response

This manuscript is written for NGS Panel in Clinical Practice for Therapy and Risk Assessment of Ductal Pancreas Carcinomas. But this manuscript was sometimes reported and not very remarkable. The authors should mention strongly the new point in the paper/abstract. Although the authors performed the NGS analysis, Sanger sequencing MLPA method, FISH, and immunohistochemical analysis, it is not always necessary and not practical. To do multiple method might be spoiled the benefits of the NGS (If the authors have shown the utility of the NGS analysis, you should modify the abstract.). 

The authors revised the abstract and improve the use of NGS analysis.

The authors described "Hystological (spelling errors?) grade was G2 in 75 13 cases and G3 in 8 cases" in "Materials and Methods", and what dose is mean? Does it have something to do with? And what is G2/G3? The pancreatic adenocarcinoma is typically classified well/moderately/poorly in the histological findings.

We modified grading classification as suggested by the reviewer.

Reviewer 2 Report

In this manuscript, Tibiletti et al. reported analysis results of 21 FFPE PDAC samples using a strategic NGS panel of 37 genes including possible PDAC related targets for therapies and cancer susceptibility genes. This study showed the potential application value of the NGS panel in both therapy choice and risk assessment for PDAC patients. However, the clinical value of this finding should be further verified.

Major concerns:

  1. A flow chart is recommended to show the overall methodology in the Materials and Methods part before being described in detail.
  2. The authors used Sanger sequencing to investigate the germline condition of actionable variants in 8 non-neoplastic tissues. The authors should perform NGS analysis in at least 2 samples to demonstrate Sanger sequencing results are compatible with NGS results.
  3. In the 3.3 FISH results part, the authors claimed that they performed FISH analysis on several PDAC samples for ERBB2, HER2, TOP2A genes. However, they only showed a FISH image for the HER2 gene in one case. I suggest the author provide more FISH imaging results as a supplementary figure. Each FISH image should include a scale bar. In addition, FISH analysis is not proper for presenting in Figure 2, because it was cited after Figure 3.
  4. In the 3.4 IHC results part, the author said they conducted IHC for Her2 and MMR, but did not show any imaging results. I also suggest the author add more IHC images in both main figures and supplementary figures.
  5. The authors failed to establish a reasonable and mechanistic link between the proposed 37 genes NGS panel with PDAC treatment and prognosis. The authors should examine the influence of identified variants in survival rate of PDAC patients, at least in public transcriptome resources of human to clarify their importance in PDAC.

Minor concerns:

  1. The resolution of the figures failed to reach the publication requirements.
  2. Abbreviations such as NGS and FFPE were not properly annotated.

Author Response

We thank for comments and suggestions that allow to improve our paper. Our revision takes into account all the points raised by Reviewers and the editorial requirements. We read and corrected writing errors.  Our point-by-point responses are listed below.

REVIEWER 2

In this manuscript, Tibiletti et al. reported analysis results of 21 FFPE PDAC samples using a strategic NGS panel of 37 genes including possible PDAC related targets for therapies and cancer susceptibility genes. This study showed the potential application value of the NGS panel in both therapy choice and risk assessment for PDAC patients. However, the clinical value of this finding should be further verified.

 Major concerns:

1.A flow chart is recommended to show the overall methodology in the Materials and Methods part before being described in detail.

We added a flow-chart describing the overall methodology used in this study (see Figures Materials and Methods).

2.The authors used Sanger sequencing to investigate the germline condition of actionable variants in 8 non-neoplastic tissues. The authors should perform NGS analysis in at least 2 samples to demonstrate Sanger sequencing results are compatible with NGS results.

Regard this point, we are not able to performe NGS analysis on FFPE samples of non-neoplastic tissues. Unfortunatly, the amount of DNA obtained from these FFPE specimens is very poor and not adequate for NGS analysis. In addition the majority of PDAC patients were dead and in all cases but one is not possible to perform constitutional analysis.

In the 3.3 FISH results part, the authors claimed that they performed FISH analysis on several PDAC samples for ERBB2, HER2, TOP2A genes. However, they only showed a FISH image for the HER2 gene in one case. I suggest the author provide more FISH imaging results as a supplementary figure. Each FISH image should include a scale bar. In addition, FISH analysis is not proper for presenting in Figure 2, because it was cited after Figure 3.

As suggested by the reviewer we added two supplementary figures showing HER2 immunohistochemical expression and FISH of PA-4, PA-1 and PA-3 cases (Figure S2). In addition we added as supplementary figure S1 showing dual color FISH for HER2, and TOP2A of PA-4 case.

In the 3.4 IHC results part, the author said they conducted IHC for Her2 and MMR, but did not show any imaging results. I also suggest the author add more IHC images in both main figures and supplementary figures.

We added immunohistochemical pictures of normal expression of MMR genes in PA-12 PDAC case as supplementary figure 3 (S3).

The authors failed to establish a reasonable and mechanistic link between the proposed 37 genes NGS panel with PDAC treatment and prognosis. The authors should examine the influence of identified variants in survival rate of PDAC patients, at least in public transcriptome resources of human to clarify their importance in PDAC.

This is a retrospective study and all PDAC patients were died, survival data for these patients are not known. In addition, the prognosis was not a goal of this work. The purpose of this work is mainly to develop an easy and reproducible test to evaluate genes involved in prognosis and therapeutic choices. For this reason and subsequently to comments of reviewer 3 we decide to change also the title of this paper.

 Minor concerns:

The resolution of the figures failed to reach the publication requirements.

 Figure 3 (now renamed as Figure 4) has been substituted with one with a better resolution.

Abbreviations such as NGS and FFPE were not properly annotated. Ok we corrected in Introduction section

Considering the possible impact of our results for clinical practice, we hope that our manuscript will be considered for publication in the prominent Biomedicines Topical Collection: Hereditary Gastrointestinal Cancer Syndromes: Molecular Basis of Onset and Progression, Molecular Diagnosis, and Precision Therapy.  Guest Editors: Prof. Marina De Rosa.

Reviewer 3 Report

In this manuscript by Tibiletti et al., authors investigate the mutational profiles of archived pancreatic ductal adenocarcinomas (n=21) using a targeted NGS panel representing 37 genes, which includes representation of genes associated with germline cancer predisposition. Using this platform, this analysis confirms the existing knowledge that PDACs contain recurrent mutations in genes with functional roles in homologous recombination and in MMR, both of which provide rationale for targeted therapies (PARPi and immune therapy, respectively). 

Aside from applying a different targeted platform, which itself is quite limited in the number of genes being evaluated, the overall impact of this manuscript and what it contributes to the field is limited. Comprehensive genomic analyses of pancreatic cancers have been published, as have custom capture panels (such as MSK-IMPACT). The associations between specific mutations and potential therapies are also not new. 

In addition to these overall concerns about novelty (which are highlighted in lines 296-300 of this manuscript), I also have several specific comments. 

  1. Line 285: What is the significance of the MMR mutations if not evidence of microsatellite instability was found? Would these tumors be expected to respond to specific therapies? How does this then impact clinical practice and decision making?
  2. The title is too vague (reads like a review article title) and should be written to emphasize the new knowledge provided by this work. Section titles in the results should also reflect more than just the technique that was used for the experiments/analysis. 
  3. The manuscript overall requires editing. Acronyms are used before they are defined (ie. Line 12, PC; Table 1 HGVS class, LIne 234 MLPA). The convention for human gene symbols is capitalized and italics. English language edits are also required (examples: line 97, line 342). In lines 246-252 authors switch between ERRB2 and HER2/Her2 a lot and this section would benefit from consistency. There are references missing (example, lines 315-316). 
  4. Table 1: in PMS2 column: what are N -/+ and Cit+?
  5. Figure 2A would benefit from including a locus that is not amplified for comparison. The text indicated TOP2A is a nearby gene that is not amplified, this would be a good example to show. 
  6. Figure 3 is generally pretty low resolution, and I would suggest that the highlighted area in 3A be blow-up and included as an inlay in the figure so this region is easier to see. 
  7. The words interestingly and interesting are over used and often refer to existing knowledge. For example line 305.
  8. Line 375: No funding for this work was acknowledged

Author Response

We thank for comments and suggestions that allow to improve our paper. Our revision takes into account all the points raised by Reviewers and the editorial requirements. We read and corrected writing errors.  Our point-by-point responses are listed  below.

REVIEWER 3

In this manuscript by Tibiletti et al., authors investigate the mutational profiles of archived pancreatic ductal adenocarcinomas (n=21) using a targeted NGS panel representing 37 genes, which includes representation of genes associated with germline cancer predisposition. Using this platform, this analysis confirms the existing knowledge that PDACs contain recurrent mutations in genes with functional roles in homologous recombination and in MMR, both of which provide rationale for targeted therapies (PARPi and immune therapy, respectively). 

Aside from applying a different targeted platform, which itself is quite limited in the number of genes being evaluated, the overall impact of this manuscript and what it contributes to the field is limited. Comprehensive genomic analyses of pancreatic cancers have been published, as have custom capture panels (such as MSK-IMPACT). The associations between specific mutations and potential therapies are also not new. 

In addition to these overall concerns about novelty (which are highlighted in lines 296-300 of this manuscript), I also have several specific comments. 

Line 285: What is the significance of the MMR mutations if not evidence of microsatellite instability was found? Would these tumors be expected to respond to specific therapies? How does this then impact clinical practice and decision making? The presence of heterozygous MMR mutation in PDAC without evidence of microsatellite instability and defect of MMR gene expression indicate that this mutations are single hits in PDAC without alterations of MMR function. Moreover MMR genes are tumor suppressor genes and the loss of function is usually due to biallelic inactivation. We think that these tumors do not respond to immunotherapies, but no data are available from literature at the moment.

The title is too vague (reads like a review article title) and should be written to emphasize the new knowledge provided by this work. Section titles in the results should also reflect more than just the technique that was used for the experiments/analysis. As suggested by the reviewer 3 we change the title of this paper as follow . “OncoPan: an NGS - based screening methodology to identify molecular markers for Therapy and Risk Assessment in Ductal Pancreas Carcinomas “

The manuscript overall requires editing. Acronyms are used before they are defined (ie. Line 12, PC; Table 1 HGVS class, LIne 234 MLPA). The convention for human gene symbols is capitalized and italics. English language edits are also required (examples: line 97, line 342). In lines 246-252 authors switch between ERRB2 and HER2/Her2 a lot and this section would benefit from consistency. There are references missing (example, lines 315-316). All acronims were checked, the references were using Endnote software. The entire paper was reviewed by a native English speaker (Mrs. Claire Thomas).

Table 1: in PMS2 column: what are N -/+ and Cit+?  This peculiar case was explained in results section

Figure 2A would benefit from including a locus that is not amplified for comparison. The text indicated TOP2A is a nearby gene that is not amplified, this would be a good example to show. We added in Supplementary S1 a detailed image of FISH analysis demonstrating in the same experiment HER2 amplkification and TOP2A not amplified.

Figure 3 is generally pretty low resolution, and I would suggest that the highlighted area in 3A be blow-up and included as an inlay in the figure so this region is easier to see. Figure 3 has been substituted with one with a better resolution.

The words interestingly and interesting are over used and often refer to existing knowledge. For example line 305. We changed.

Line 375: No funding for this work was acknowledged. It is correct

Considering the possible impact of our results for clinical practice, we hope that our manuscript will be considered for publication in the prominent Biomedicines Topical Collection: Hereditary Gastrointestinal Cancer Syndromes: Molecular Basis of Onset and Progression, Molecular Diagnosis, and Precision Therapy.  Guest Editors: Prof. Marina De Rosa.

Round 2

Reviewer 1 Report

OK

Reviewer 2 Report

My comments have been appropriately addressed in the revised manuscript and I would like to recommend its publication.

Reviewer 3 Report

While I do believe the revised manuscript is improved over the original version, the conceptual novelty and significance of this work remain limited.

A few specific comments are provided below:

-Figure 4 is better resolution but still very hard to read. For a) I would recommend adding and inlay of the regions indicated with the arrow at a larger size so that it can be read more easily; for b) there is too much white space and the axes could be adjusted to make the panel more readable. ­

- I appreciate the additional of the FISH data for TOP2A, however, since the foundation and basis of this paper is NGS, I would like to see the data from the NGS included as well (like Figure 3A).    

-Genes symbols should be italicized throughout the manuscript and some additional English language edits are required. For example – line 102: “For NGS analysis it was designed a custom hybrid capture-based…”. I suggest: “For NGS analysis, a custom hybrid capture-based panel was designed and named OncoPan”.